# Comprehensive interrogation of the ADAR2 deaminase domain for engineering enhanced RNA editing activity and specificity

Dhruva Katrekar[1], Yichen Xiang[1], Nathan Palmer[2], Anushka Saha[1], Dario Meluzzi[1], Prashant Mali[1]*

[1]Department of Bioengineering, University of California San Diego, San Diego, United States; [2]Division of Biological Sciences, University of California San Diego, San Diego, United States

**Abstract** Adenosine deaminases acting on RNA (ADARs) can be repurposed to enable programmable RNA editing, however their exogenous delivery leads to transcriptome-wide off-targeting, and additionally, enzymatic activity on certain RNA motifs, especially those flanked by a 5′ guanosine is very low thus limiting their utility as a transcriptome engineering toolset. Towards addressing these issues, we first performed a novel deep mutational scan of the ADAR2 deaminase domain, directly measuring the impact of every amino acid substitution across 261 residues, on RNA editing. This enabled us to create a domain-wide mutagenesis map while also revealing a novel hyperactive variant with improved enzymatic activity at *5′-GAN-3′* motifs. As overexpression of ADAR enzymes, especially hyperactive variants, can lead to significant transcriptome-wide off-targeting, we next engineered a split-ADAR2 deaminase which resulted in >100-fold more specific RNA editing as compared to full-length deaminase overexpression. Taken together, we anticipate this systematic engineering of the ADAR2 deaminase domain will enable broader utility of the ADAR toolset for RNA biotechnology applications.

*For correspondence:
pmali@ucsd.edu

## Editor's evaluation

This manuscript provides a deep mutational scanning of the deaminase domain of human ADAR2 to provide a comprehensive assessment of amino acids that alter editing activity at a specific adenosine flanked by preferred nucleotides (UAG). The results are quite important in terms of impact on precision medicine.

## Introduction

Adenosine to inosine (A-to-I) editing is a common post-transcriptional modification in RNA that occurs in a variety of organisms, including humans. This A-to-I deamination of specific adenosines in double-stranded RNA is catalyzed by enzymes called adenosine deaminases acting on RNA (ADARs) (*Melcher et al., 1996*; *Bass and Weintraub, 1988*; *Peng et al., 2012*; *Nishikura, 2016*; *Eggington et al., 2011*; *Wagner et al., 1989*; *Bass and Weintraub, 1987*; *Mannion et al., 2014*; *Tan et al., 2017*; *Tomaselli et al., 2015*; *Levanon et al., 2004*; *Schoft et al., 2007*). Since inosine is structurally similar to guanosine, it is interpreted as a guanosine during the cellular processes of translation and splicing, thereby making ADARs powerful systems for altering protein sequences.

Correspondingly, adenosine deaminases have been repurposed for site-specific RNA editing by recruiting them to target RNA sequences using engineered ADAR-recruiting RNAs (adRNAs) (*Woolf et al., 1995*). Recently, several studies have demonstrated the potential of both genetically encodable and chemically modified RNA-guided adenosine deaminases for the correction of point mutations and the repair of premature stop codons both in vitro (*Montiel-Gonzalez et al., 2013*; *Stafforst and Schneider, 2012*; *Cox et al., 2017*; *Wettengel et al., 2017*; *Merkle et al., 2019*; *Sinnamon et al., 2017*; *Monteleone et al., 2019*; *Fukuda et al., 2017*; *Qu et al., 2019*) and in vivo (*Katrekar et al., 2019*; *Sinnamon et al., 2020*). These studies have primarily relied on exogenous ADARs which introduce a significant number of transcriptome-wide off-target A-to-I edits (*Cox et al., 2017*; *Katrekar et al., 2019*; *Vallecillo-Viejo et al., 2018*; *Vogel et al., 2018*). One solution to this problem is the engineering of adRNAs to enable the recruitment of endogenous ADARs. In this regard, we recently showed that using simple long antisense RNA (>60 bp) can suffice to recruit endogenous ADARs and these adRNAs are both genetically encodable and chemically synthesizable (*Katrekar et al., 2019*); and Merkle and colleagues showed that using engineered chemically synthesized antisense oligonucleotides (*Merkle et al., 2019*) could also lead to robust RNA editing via endogenous ADAR recruitment. Although this modality allows for highly specific editing, its applicability is restricted to editing adenosines in certain RNA motifs preferred by the native ADARs, and in tissues with high endogenous ADAR activity. Additionally, it cannot be utilized for novel functionalities such as deamination of cytosine to uracil (C-to-U) editing which requires exogenous delivery of ADAR2 variants (*Abudayyeh et al., 2019*). Thus, engineering a genetically encodable RNA editing tool that efficiently edits RNA with high specificity and activity is essential for enabling broader use of this toolset for biotechnology and therapeutic applications.

In this regard, the crystal structure of the ADAR2 deaminase domain (ADAR2-DD) (*Macbeth et al., 2005*; *Matthews et al., 2016*; *Thuy-Boun et al., 2020*) and several key biochemical and computational studies (*Ohman et al., 2000*; *Kuttan and Bass, 2012*; *Daniel et al., 2017*; *Daniel et al., 2012*; *Dawson et al., 2004*; *Wang and Beal, 2016*; *Roth et al., 2019*; *Schaffer et al., 2020*; *Stefl et al., 2010*; *Riedmann et al., 2008*) have laid the foundation for understanding its catalytic mechanism and target preferences, but we still lack comprehensive knowledge of how mutations and fragmentation affect the ability of the ADAR2-DD to edit RNA. To address this, we first carried out a quantitative deep mutational scan (DMS), measuring the effect of every possible single amino acid substitution across 261 residues of the ADAR2-DD, on enzyme function. We utilized the sequence-function map thus generated, to identify novel enhanced variants for A-to-I editing. Additionally, combining information from these sequence-function maps with existing knowledge of the structure and residue conservation scores, we also engineered a genetically encodable split-ADAR2 system that enabled efficient and highly specific RNA editing.

## Results

### DMS of the ADAR2-DD

To gain comprehensive insight into how mutations affect the ADAR2-DD, we used DMS, a technique that enables simultaneous assessment of the activities of thousands of protein variants (*Fowler and Fields, 2014*; *Araya and Fowler, 2011*). Typically, this approach relies on phenotypic selection methods such as cell fitness or fluorescent reporters that result in an enrichment of beneficial variants and a depletion of deleterious variants. However, as RNA editing yields are not precisely quantifiable using surrogate readouts, we focused on directly measuring RNA editing activity in the screens. To do so, we linked genotype to phenotype by placing the RNA editing sites on the same transcript encoding the deaminase variant. These RNA editing sites were chosen within the deaminase domain such that they met two criteria: (1) the sites were outside the 261 residue region where single amino acid substitutions were created for the DMS; and (2) an A-to-I(G) change at these sites resulted in a synonymous alteration at the codon level. By ensuring every cell in the pooled screen received a single library element, we could perform a quantitative DMS of the core 261 amino acids (residues 340–600) of the ADAR2-DD via 4959 (261 × 19) single amino acid variants, directly measuring the effect of each mutation on A-to-I editing yields (*Figure 1a, b, and c*).

Given the large size of the deaminase domain at >750 bp, the library of ADAR variants was created using six tiling oligonucleotide pools (*Figure 1—figure supplement 1a*). These pools were cloned

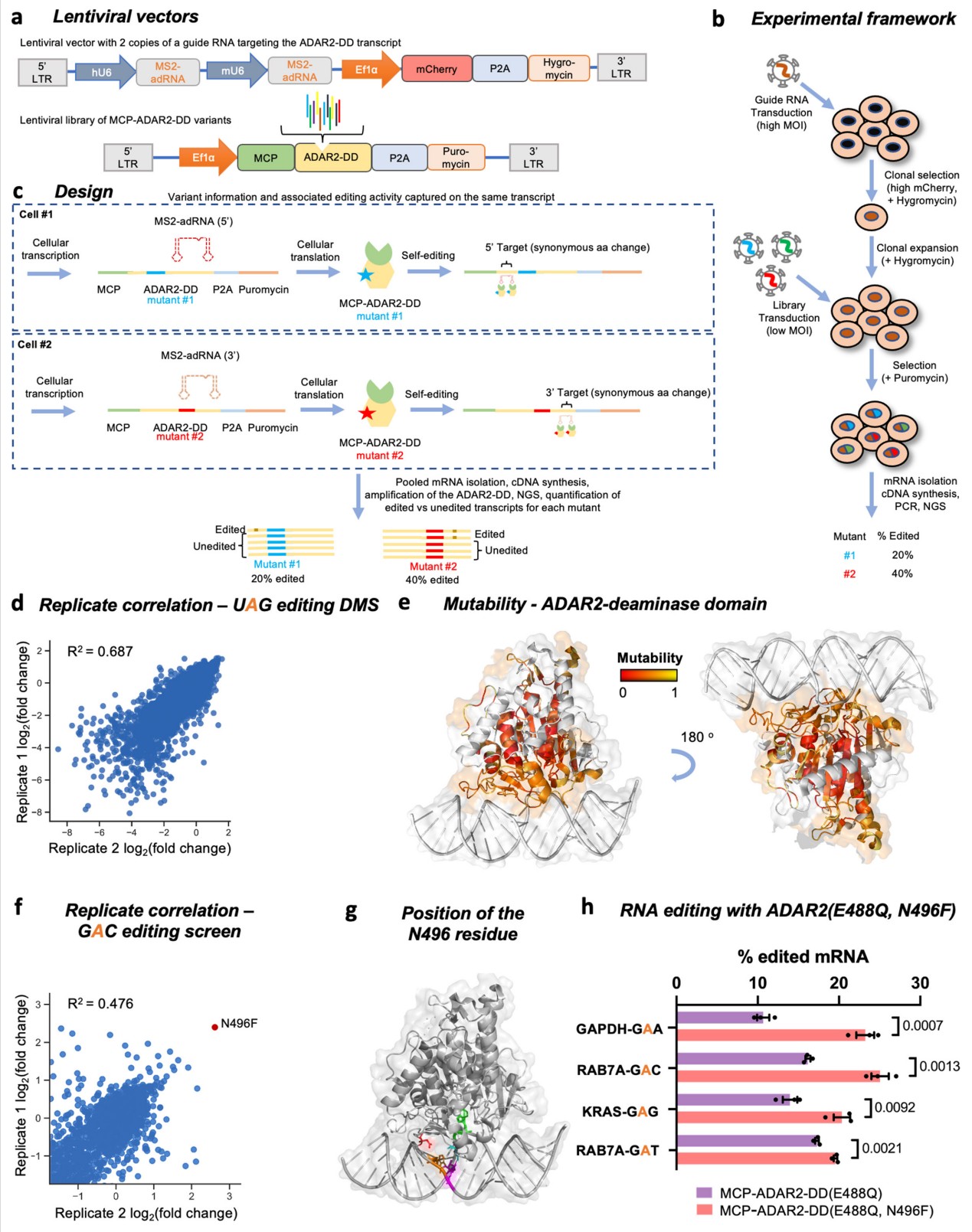

**Figure 1.** Overview of the ADAR2 deaminase domain (ADAR2-DD) deep mutational scan (DMS) and *5′-GAN-3′* enhancer editing screen. (**a**) Lentiviral vectors comprising two copies of MS2-adRNA both targeting either a 5′ or a 3′ UAG target site for the DMS were created. For the *5′-GAN-3′* enhancer screen, the MS2-adRNAs targeted a 5′ or 3′ GAC site. The lentiviral vector alco contained a mCherry and hygromycin resistance marker. Libraries of single amino acid variants of the ADAR2-DD were created in a second lentiviral with a puromycin selection marker. For the *5′-GAN-3′* enhancer screen,

*Figure 1 continued on next page*

*Figure 1 continued*

the corresponding library was created in the hyperactive ADAR2-DD(E488Q) backbone. (**b**) HEK293FT cells were transduced with the MS2-adRNA lentiviruses at a high multiplicity of infection (MOI) and upon hygromycin selection, a single clone with high mCherry expression was selected. Four independent clonal cell lines were created, harboring MS2-adRNA targeting the 5' and 3' UAG and GAC sites. The clonal cell line bearing the MS2-adRNA was then transduced with the lentiviral library of MCP-ADAR2-DD variants at a low MOI to ensure delivery of a single variant per cell. Cells were then selected with puromycin. (**c**) During the processes of cellular transcription and translation, each cell produces the MS2-adRNA as well as MCP-ADAR2-DD variant. Upon translation in the cell, each MCP-ADAR2-DD variant, in combination with the MS2-adRNA, edits its own transcript creating a synonymous change. Cells were harvested, mRNA isolated and regions of the ADAR2-DD were amplified and sequenced. The fraction of edited reads was then computed for each mutant. (**d**) Replicate correlation for the ADAR2-DD DMS. The X and Y axes represent the log2 fold change in editing as compared to the ADAR2. (**e**) Structure of the ADAR2-DD bound to its substrate (PDB 5HP3) with the degree of mutability of each residue as measured by the DMS highlighted. Residues that are highly intolerant to mutations are colored red while residues that are highly mutable are colored yellow. Residues not assayed in this DMS are colored white. (**f**) Using the library chassis of the DMS, a screen of deaminase domain mutants (in an E488Q background) was performed to mine variants with improved activity against *5'-GAN-3'* RNA motifs. Replicate correlation for the *5'-GAN-3'* enhancer mutant screen. The X and Y axes represent the log2 fold change in editing as compared to the ADAR2. (**g**) Structure of the ADAR2-DD(E488Q) bound to its substrate (PDB 5ED1) with the N496 residue highlighted in red, the E488Q residue in cyan, the target adenosine in green, the orphaned cytosine in magenta, and the adenosine on the unedited strand that base pairs with the 5' uracil flanking the target adenosine in orange. (**h**) The E488Q, N496F double mutant was validated by editing a GAA motif in the GAPDH CDS, a GAG motif in the KRAS CDS and GAC, GAT motifs in the RAB7A 3' UTR. Values represent mean ± SEM (n = 3). p-Values were computed using a two-tailed unpaired t-test. All experiments were carried out in HEK293FT cells.

The online version of this article includes the following source data and figure supplement(s) for figure 1:

**Source data 1.** ADAR2 deaminase domain deep mutational scan and *5'-GAN-3'* enhancer editing screen.

**Figure supplement 1.** Deep mutational scan (DMS) of the ADAR2 deaminase domain (ADAR2-DD).

**Figure supplement 2.** Characterization of ADAR2 deaminase domain (ADAR2-DD) mutants.

**Figure supplement 3.** Characterization of ADAR2 deaminase domain (ADAR2-DD) E488Q, N496F mutant.

into a lentiviral vector containing a sequence encoding the MS2 coat protein (MCP) and the remainder of the deaminase domain bearing a nuclear export signal (NES) followed by a self-cleaving peptide (P2A) and a puromycin resistance gene (**Figure 1a**). To ensure read length coverage in next generation sequencing, members of the first three library pools were assayed for editing at the 5' end of the ADAR2-DD while the remaining members were assayed at the 3' end (**Figure 1c**, **Figure 1—figure supplement 1a**). To minimize assay noise resulting from varying expression of the guide RNA in each cell, two HEK293FT clonal cell lines were created following a high multiplicity of infection (MOI) transduction with lentivirus bearing two copies of MS2-adRNAs targeting either the 5' or the 3' UAG sites integrated into them (**Figure 1b**). The DMS was carried out in cell lines harboring these MS2-adRNAs by transducing them with the corresponding ADAR mutant libraries at a low MOI of 0.2–0.4. Two biological replicates were performed in independent plates of cells transduced with the lentiviral libraries. Following lentiviral transduction and puromycin selection, RNA was extracted from the harvested cells and reverse transcribed. Relevant regions of the deaminase domain were amplified from the cDNA and sequenced (**Figure 1—figure supplement 1a**); 4958 of the 4959 possible variants were successfully detected, of which 4931 elements had over 50 reads in each biological replicate and were included in the subsequent analysis (**Figure 1—figure supplement 1b**). The deaminase domain transcripts for each variant also contained the associated A-to-I editing yields, which were then quantified for both replicates of the DMS ($R^2 = 0.687$) (**Figure 1d**).

The scans revealed both intrinsic domain properties, and also several mutations that enhanced RNA editing (**Figure 1e**, **Figure 1—figure supplement 1c**). Specifically: (1) As expected, most mutations in conserved regions 442–460 and 469–495 that bind the RNA duplex near the editing site led to a significant decrease in editing efficiency of the enzyme (**Matthews et al., 2016**). (2) However, mutating the negatively charged E488 residue, which recognizes the cytosine opposite the flipped adenosine by donating hydrogen bonds, to a positively charged or most polar-neutral amino acids resulted in an improvement in editing efficiency. This is consistent with the previously discovered E488Q mutation which has been shown to improve the catalytic activity of the enzyme (**Kuttan and Bass, 2012**). (3) Furthermore, most mutations to residues that contact the flipped adenosine (V351, T375, K376, E396, C451, R455) were observed to be detrimental to enzyme function (**Matthews et al., 2016**). (4) Similarly, the residues of the ADAR2-DD that interact with the zinc ion in the active site and the inositol hexakisphosphate (R400, R401, K519, R522, S531, W523, D392, K483, C451, C516, H394, and E396) were all also extremely intolerant to mutations (**Macbeth et al., 2005**). (5) Additionally, as

expected, surface exposed residues in general readily tolerated mutations as compared to buried residues (*Matthews et al., 2016*).

To independently validate the results from the DMS, we individually examined 33 mutants from the DMS whose editing efficiencies ranged from very low to very high as compared to the wild-type ADAR2-DD. The mutants were assayed for their ability to repair a premature amber stop codon (UAG) in the *cypridina* luciferase (cluc) transcript (*Cox et al., 2017*). The Pearson correlation between the arrayed validations and the data obtained in the screen was 0.818 while the Spearman (rank) correlation was 0.824 (*Figure 1—figure supplement 2a*). Additionally, we also validated several of these mutants for their ability to edit UAG motifs in the GAPDH and KRAS CDS (*Figure 1—figure supplement 2b*). These validations suggest that the trends in RNA editing activity (high vs. low) are well predicted using this novel screening approach and this enabled us to create a mutability map of the ADAR2-DD where residues that tolerate mutations are highlighted in yellow (*Figure 1e*). Additionally, we compared the efficiency of variants in our ADAR2-DD DMS at editing UAG triplets, to published mutants (*Matthews et al., 2016*; *Thuy-Boun et al., 2020*; *Kuttan and Bass, 2012*) and again observed similar agreement in the trends of activity of most variants, confirming the efficacy of this DMS at accurately indicating whether a mutation is beneficial, neutral, or detrimental for enzymatic activity.

## Enhancing enzyme activity at *5'-GAN-3'* motifs

Building on this platform (*Figure 1a, b, and c*), we next screened for domain variants that improved editing at refractory RNA motifs such as adenosines flanked by a 5' guanosine (*Vogel et al., 2018*; *Kuttan and Bass, 2012*). Toward this, two HEK293FT clonal cell lines were created with two copies of MS2-adRNAs targeting either the 5' or 3' GAC sites integrated into them. A screen was carried out in cell lines harboring these MS2-adRNAs by transducing them with the corresponding MCP-ADAR2-DD(E488Q) libraries at a low MOI (0.2–0.4), evaluating the potential of 3287 mutants to edit a GAC motif. Similar to above, following lentiviral transduction and selection, RNA was extracted, reverse transcribed, and relevant regions of the deaminase domain amplified, sequenced, and analyzed ($R^2$ = 0.476) (*Figure 1f*). Via this approach, we discovered a novel mutant E488Q, N496F that enhanced editing at a *5'-GAC-3'* motif. Interestingly, in the ADAR2-DD(E488Q) crystal structure, the N496 residue is in close proximity to the adenosine on the unedited strand that base pairs with the 5' uracil flanking the target adenosine (*Figure 1g*; *Matthews et al., 2016*). We validated this mutant using a *cypridina* luciferase (cluc) reporter bearing a premature opal stop codon (UGA) and confirmed that the E488Q, N496F double mutant was 3-fold better at restoring luciferase activity as compared to E488Q alone (*Figure 1—figure supplement 3a*). To further confirm that the E488Q, N496F double mutant could be used to efficiently edit adenosines flanked by a 5' guanosine, we tested the ability of this mutant to edit GAC, GAT, GAG, and GAA motifs in the endogenous RAB7A (3' UTR), KRAS (CDS), and GAPDH (CDS) transcripts. We observed that the double mutant E488Q, N496F was 1.1- to 2.1-fold more efficient at editing various *5'-GAN-3'* motifs as compared to the E488Q (*Figure 1h*, *Figure 1—figure supplement 3b and c*), confirming the ability of this novel screening format to discover variants with enhanced activity toward refractory RNA motifs. We also confirmed that this new variant was at least as efficient as the E488Q at editing all other motifs, with improved editing also observed at *5'-NAC-3'* motifs (*Figure 1—figure supplement 3c*).

## Improving specificity via splitting of the ADAR2-DD

In addition to increasing the on-target activity of ADARs at editing adenosines in non-preferred motifs, another challenge toward unlocking their utility as a RNA editing toolset is that of improving specificity. Due to their intrinsic dsRNA binding activity, overexpression of ADARs leads to promiscuous transcriptome-wide off-targeting, and thus, when relying on exogenous ADARs, it is important to restrict the catalytic activity of the overexpressed enzyme only to the target mRNA. We hypothesized that it might be possible to achieve this by splitting the deaminase domain into two catalytically inactive fragments that come together to form a catalytically active enzyme only at the intended target (*Figure 2a*). Since we and others have utilized the MCP and Lambda N ($\lambda$N) systems to efficiently recruit ADARs, we first decided to utilize these systems to recruit the two split halves, that is, the N- and C-terminal fragments of the ADAR2-DD (*Montiel-Gonzalez et al., 2013*; *Katrekar et al., 2019*). Specifically, constructs were created with cloning sites for N-terminal fragments located downstream of the MCP while those for the C-terminal fragments located upstream of the $\lambda$N. Chimeric adRNAs

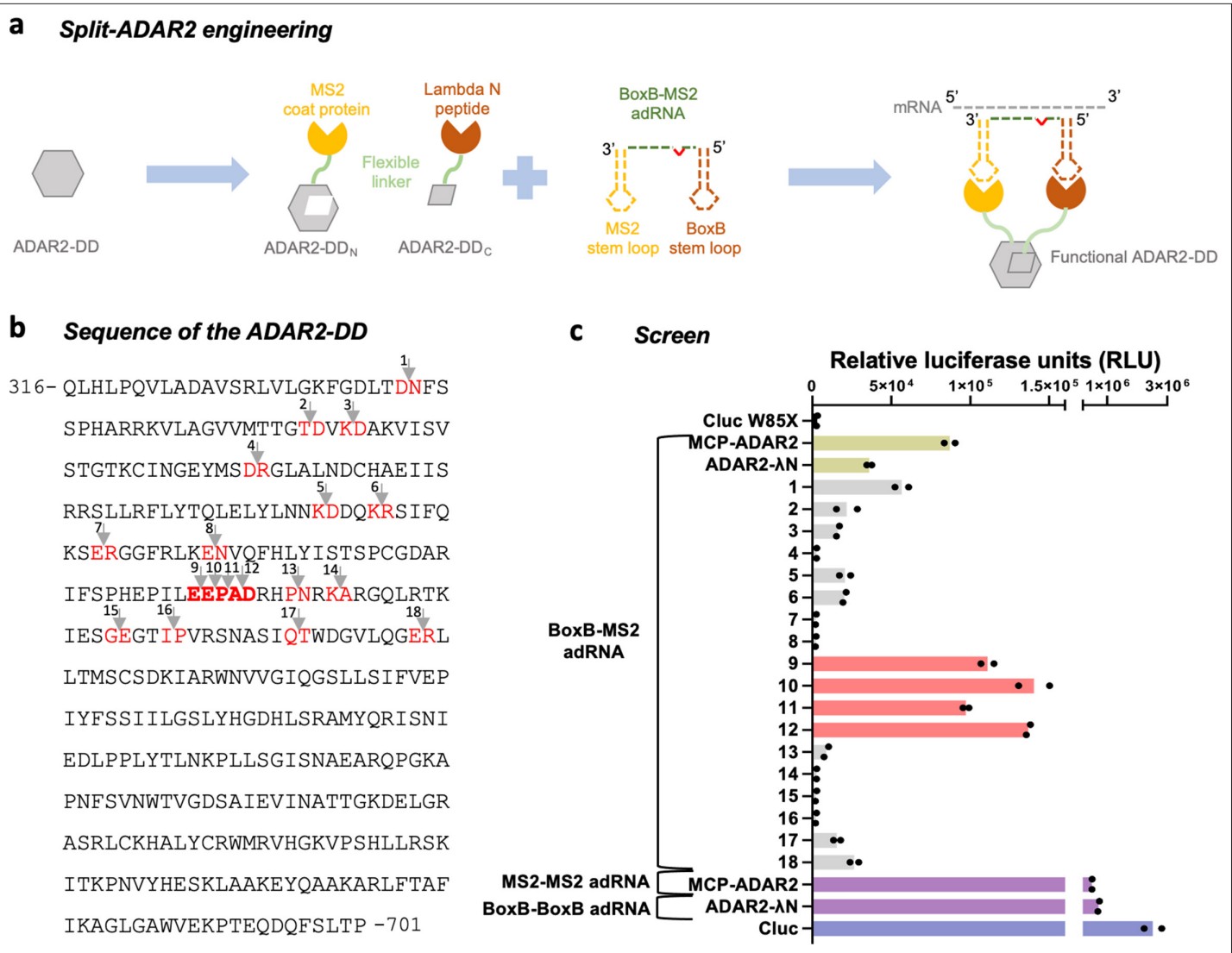

**Figure 2.** Engineering split-ADAR2 deaminase domains (ADAR2-DD). (**a**) Schematic of the split-ADAR2 engineering approach. (**b**) Sequence of the ADAR2-DD. The protein was split between residues labelled in red, and a total of 18 pairs were evaluated. (**c**) The ability of each split pair from (**b**) to correct a premature stop codon when transfected with a chimeric BoxB-MS2 ADAR-recruiting RNA (adRNA) was assayed via a luciferase assay. The pairs 1–18 correspond to the residues in red in (**b**) in the order in which they appear. The residues in (**b**) in bold red correspond to pairs 9–12. Values represent mean (n = 2). All experiments were carried out in HEK293FT cells.

The online version of this article includes the following source data and figure supplement(s) for figure 2:

**Source data 1.** Engineering split-ADAR2 deaminase domains.

**Figure supplement 1.** Optimization of the split-ADAR2 deaminase constructs.

were designed to bear a BoxB and an MS2 stem loop along with an antisense domain complementary to the target. Examining the results of the DMS (focusing on sites with high mutability), as well as the crystal structure of the ADAR2-DD (focusing on high solvent accessible surface area), and residue conservation scores across species (focusing on low scores of conservation), we identified 18 putative regions for splitting the protein (*Figure 2b*; *Matthews et al., 2016*; *Dagliyan et al., 2018*). The resulting 18 different split-ADAR2 pairs were assayed for their ability to repair a premature amber stop codon (UAG) in the *cypridina* luciferase (cluc) transcript in the presence of the recruiting adRNA bearing BoxB and MS2 stem loops (*Figure 2c*). Of these pairs 9–12 showed the best editing efficiency, and notably were all located within residues 465–468 which have low conservation scores across species (*Matthews et al., 2016*). Interestingly, this region is flanked by highly conserved amino

acids (442–460 and 469–495). The split-ADAR2 pair 12 is hereon referred to as ADAR2-DD$_N$ and ADAR2-DD$_C$.

We also confirmed that every component of the split-ADAR2 system was essential for RNA editing. Specifically, we assayed all components and pairs of components for their ability to edit the RAB7A transcript and also restore luciferase activity. The MCP-ADAR2-DD was included as a control. We observed editing of the RAB7A transcript and restoration of luciferase activity only when every component of the split-ADAR2 system was delivered, confirming that the individual components lacked enzymatic activity (*Figure 3a*, *Figure 2—figure supplement 1a*). Additionally, we also confirmed the importance of fragment orientation for the formation of a functional enzyme. Toward this, we swapped the positions of the N- and C-terminal fragments and created ADAR2-DD$_N$-MCP and $\lambda$ N-ADAR2-DD$_C$ in addition to the working MCP-ADAR2-DD$_N$ and ADAR2-DD$_C$-$\lambda$ N pair. We then tested each pair of N- and C-terminal fragments and observed functionality only for the MCP-ADAR2-DD$_N$ paired with ADAR2-DD$_C$-$\lambda$ N (*Figure 2—figure supplement 1b*).

Since MCP and $\lambda$ N are proteins of viral origin we next replaced these with the human TAR binding protein and the stem loop binding protein, respectively, to create a humanized split-ADAR2 system with improved translational relevance (*Rauch et al., 2019*). In the presence of a chimeric adRNA containing a histone stem loop and a TAR stem loop, we observed restoration of luciferase activity (*Figure 2—figure supplement 1c*). This also confirmed that the split-ADAR2-DD could indeed be recruited for RNA editing using two independent sets of protein-RNA binding systems.

Finally, we investigated the specificity profiles via analysis of the transcriptome-wide off-target A-to-I(G) editing effected by this system (*Figure 3b* and *Figure 3—figure supplements 1 and 2*). Each condition from *Figure 3a* (where a UAG in the endogenous RAB7A transcript was targeted) was analyzed by RNA-seq. From each sample, we collected 25 million uniquely aligned sequencing read pairs. We then used Fisher's exact test to quantify significant changes in A-to-G editing yields, relative to untransfected cells, at each reference adenosine site having sufficient read coverage. As expected with hyperactive enzyme variants, the ADAR2-DD(E488Q) showed significantly high off-target editing as compared to the ADAR2-DD (*Figure 3b*). Notably, by splitting the deaminase domain, we observed a 1000- to 1300-fold reduction in the number of off-targets as compared to the full-length ADAR2-DD or ADAR2-DD(E488Q) (*Figure 3b*). Excitingly, the specificity profiles of the split-ADAR2 system were comparable to those seen when using endogenous recruitment of ADARs via long antisense RNA (*Katrekar et al., 2019*).

To confirm generalizability of the results, we also tested the split-ADAR2 at two additional endogenous loci: an adenosine in the 3'UTR of CKB and an adenosine in the CDS of KRAS, and observed robust editing efficiency of the split-ADAR2 system (*Figure 3c*). To enable convenient delivery of the split-ADAR2 system, we also created an all-in-one vector bearing a bicistronic ADAR2-DD$_C$-$\lambda$ N-P2A-MCP-ADAR2-DD$_N$ which also enabled higher editing efficiencies across all three loci tested (*Figure 3a and c*). We also confirmed transcriptome-wide specificity of targeting using the all-in-one vector while targeting RAB7A (*Figure 3—figure supplement 2*) and also while targeting the KRAS locus (*Figure 3—figure supplement 3*). A closer look at the off-targets revealed that in case of the split-ADAR2 system, highly edited off-targets were indeed guide RNA sequence dependent. This is in contrast to full-length deaminase domain overexpression where off-targets were predominantly deaminase domain driven (*Supplementary file 1*). The entire split-ADAR2 system consisting of CMV promoter-driven ADAR2-DD$_C$-$\lambda$ N-P2A-MCP-ADAR2-DD$_N$ and a human U6 promoter-driven BoxB-MS2 adRNA is ~3500 bp in size and can easily be packaged into a single adeno-associated virus (AAV).

Lastly, to test if the split-ADAR2 chassis could be expanded to highly active ADAR variants that enable efficient editing of *5'-GAN-3'* motifs as well as those that enable C-to-U editing, we created a split-ADAR2-DD(E488Q, N496F) and a split-RESCUE (RNA Editing for Specific C-to-U Exchange) system (*Abudayyeh et al., 2019*). We confirmed that the split-ADAR2-DD(E488Q, N496F) was indeed better at editing a GAC motif as compared to the split-ADAR2-DD(E488Q) (*Figure 4a*). Although the full-length ADAR2-DD(E488Q, N496F) was highly promiscuous, splitting it enabled high transcriptome-wide specificity while targeting a UAG in the RAB7A transcript (*Figure 4b*, *Figure 3—figure supplement 2*). Additionally, we noted comparable C-to-U RNA editing levels of the endogenous RAB7A transcript using the split-RESCUE and the full-length MCP-RESCUE (*Figure 4c*). However,

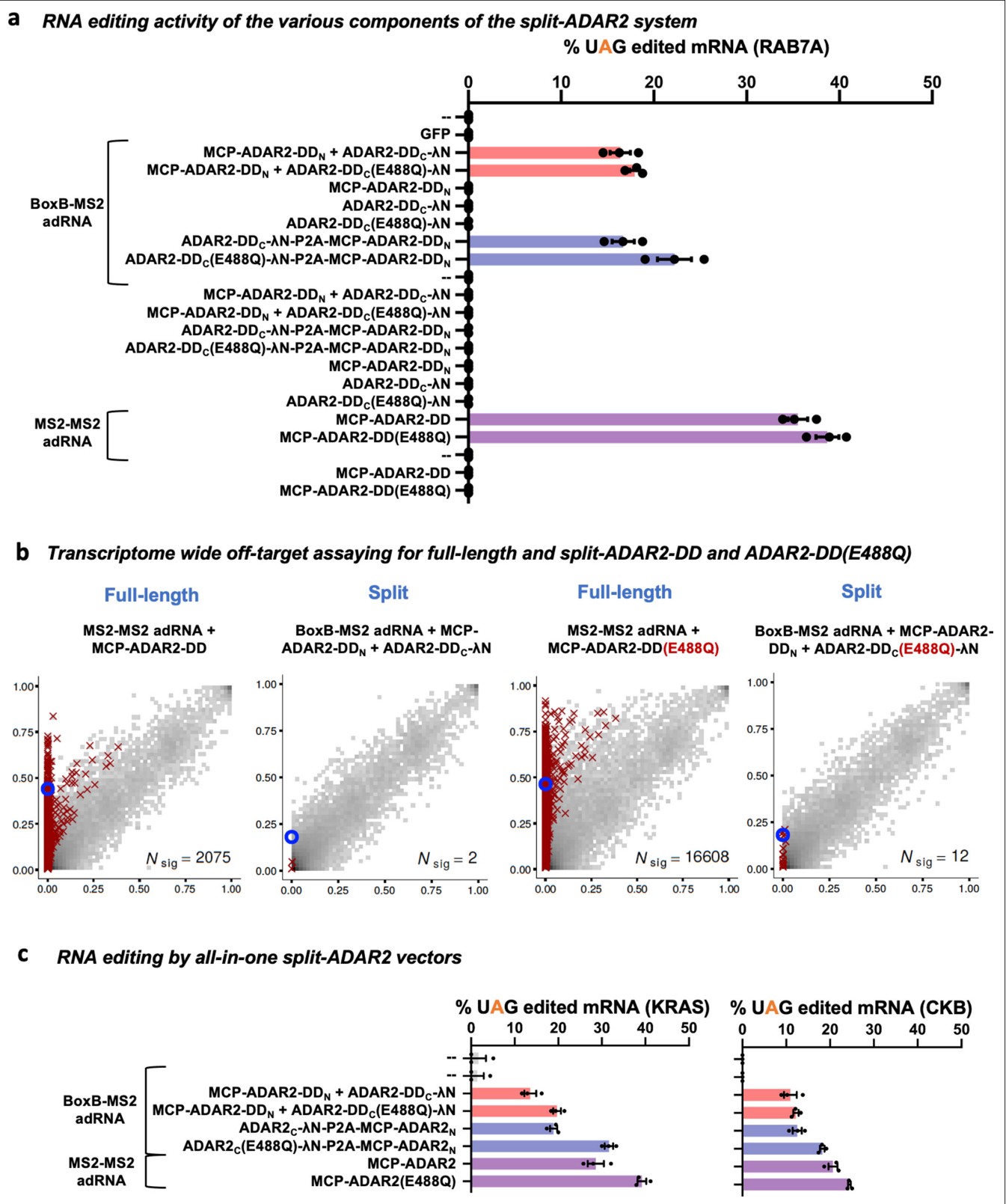

**Figure 3.** Characterizing the split-ADAR2 deaminase domains. (**a**) The components of the split-ADAR2 system based on pair 12 were tested for their ability to edit the RAB7A transcript. Editing was observed only when every component was delivered. Values represent mean ± SEM (n = 3). (**b**) 2D histograms comparing the transcriptome-wide A-to-G editing yields observed with each construct (y-axis) to the yields observed with the control sample (x-axis). Each histogram represents the same set of reference sites, where read coverage was at least 10 and at least one putative editing event

*Figure 3 continued on next page*

*Figure 3 continued*

was detected in at least one sample. Bins highlighted in red contain sites with significant changes in A-to-G editing yields when comparing treatment to control sample. Red crosses in each plot indicate the 100 sites with the smallest adjusted p-values. Blue circles indicate the intended target A-site within the RAB7A transcript. (**c**) The split-ADAR2 system was assayed for editing the KRAS and CKB transcripts. Values represent mean ± SEM (n = 3). All experiments were carried out in HEK293FT cells.

The online version of this article includes the following source data and figure supplement(s) for figure 3:

**Source data 1.** Characterizing the split-ADAR2 deaminase domains.

**Figure supplement 1.** Specificity profiles of the ADAR2 deaminase expression constructs.

**Figure supplement 2.** Specificity profiles of the ADAR2 deaminase expression constructs.

**Figure supplement 3.** Specificity profiles of the split-ADAR2 deaminase expression constructs.

the split-RESCUE system was also highly specific, both in the A-to-I(G) and C-to-U space as compared to the MCP-RESCUE (*Figure 4d*).

## Discussion

Toward addressing two of the fundamental challenges in using ADARs for programmable RNA editing, specifically, (1) poor enzymatic activity on certain RNA motifs such as those flanked by a 5' guanosine (*Vogel et al., 2018*; *Kuttan and Bass, 2012*), and (2) exogenous delivery leading to massive transcriptome-wide off-targeting (*Cox et al., 2017*; *Katrekar et al., 2019*; *Vallecillo-Viejo et al., 2018*; *Vogel et al., 2018*), we have explored ADAR2 deaminase protein engineering via two distinct approaches. First, we performed a novel DMS, comprehensively assaying all possible single amino acid substitutions of 261 residues of the deaminase domain for their impact on RNA editing yields. We created a sequence-function map of the deaminase domain that complements existing knowledge derived from prior structure and biochemistry-based studies and improves our understanding of the enzyme. This can serve as a map for engineering novel ADAR2 variants with tailored activity for specific applications. For instance, while utilizing the ADAR enzyme to create transcriptomic timestamps, tailored variants might enable capture of information at different time scales (*McMahon et al., 2016*). Additionally, use of tailored variants might help better understand kinetics of protein-RNA interactions (*Rodriques et al., 2021*). This novel screening approach also enabled us to identify variants such as the ADAR2-DD(E488Q, N496F) that demonstrated increased activity at *5'-GAN-3'* motifs. Specifically, this mutant was 1.1- to 2.1-fold more efficient at editing adenosines with a 5' guanosine than the classic hyperactive ADAR2-DD(E488Q) and also maintained similar activity levels against all other motifs. However, like the ADAR2-DD(E488Q), the ADAR2-DD(E488Q, N496F) showed increased bystander editing and transcriptome-wide off-targeting as compared to the ADAR2-DD. While this novel screening approach was useful for creating an average mutagenesis map of the ADAR2-DD and identifying highly active variants, we believe that replicate correlation values were negatively impacted given that the screen was carried out in a lentiviral format. As lentivirus integrates randomly into the cell's genome, this results in variability in ADAR2-DD transcript levels between different cells.

Second, we engineered split deaminases, consisting of two inactive enzyme fragments that formed a functional enzyme upon combining at the target site. Due to this requirement of the split-domains to assemble, the efficiency of this system was ~50–70% compared to full-length domain overexpression, but the split-ADAR2 tool was highly transcript specific (~1000-fold compared to full-length domain over expression), and notably with off-target profiles similar to those seen via recruitment of endogenous ADARs (*Katrekar et al., 2019*). We further demonstrated the applicability of these split-deaminases toward editing *5'-GAN-3'* motifs and uracils via creation of a split-ADAR2-DD(E488Q, N496F) and a split-RESCUE, both of which exhibited high transcriptome-wide specificity as compared to their full-length counterparts. In summary, this study enables broader utility of the ADAR toolset for biotechnology and therapeutic applications. Additionally, these approaches could also be applied to the study and engineering of other RNA modifying enzymes (*Rosenberg et al., 2011*; *Liu et al., 2014*).

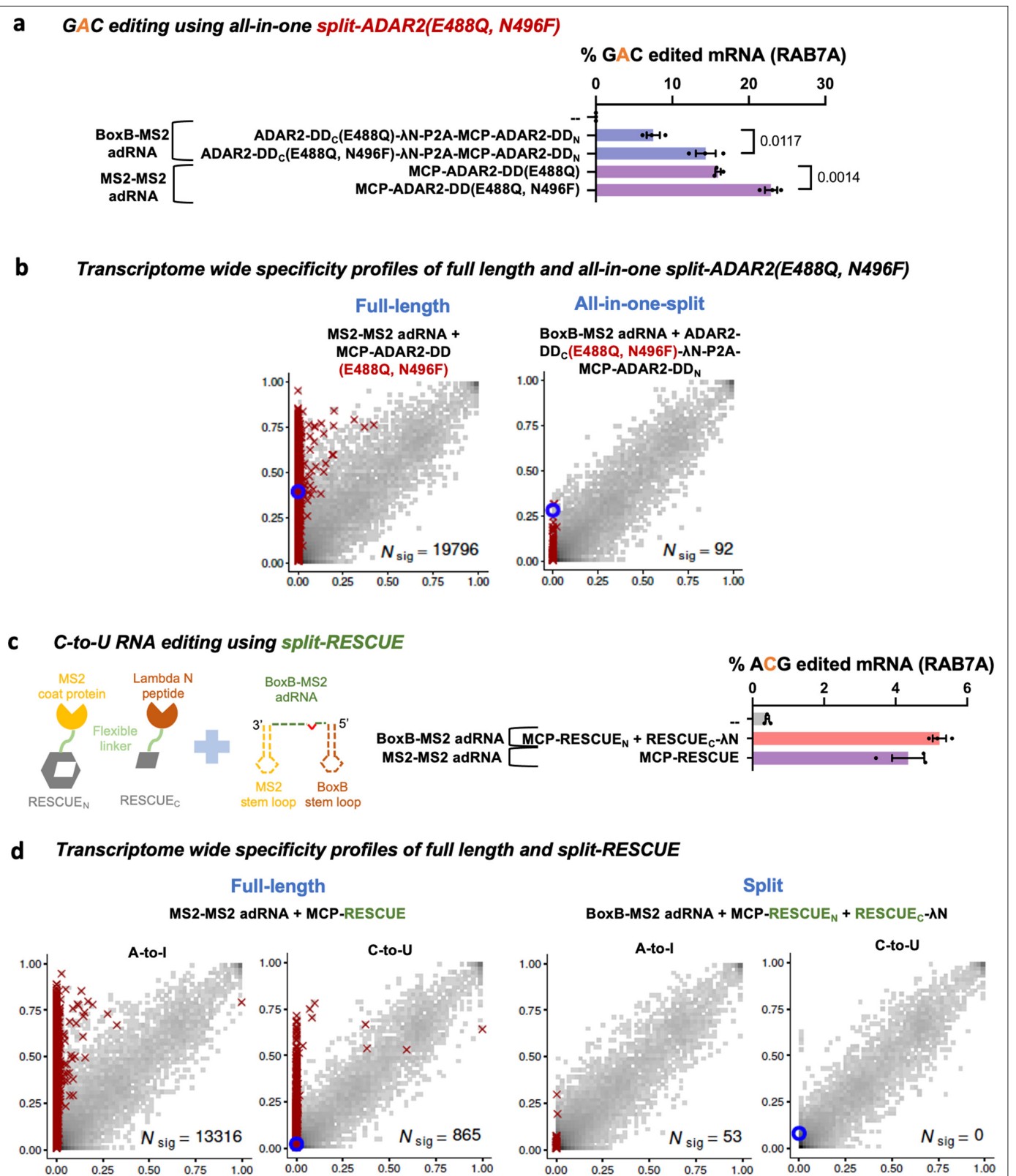

**Figure 4.** Optimizing and expanding the utility of split-ADAR2 deaminase domains (ADAR2-DD). (**a**) A split-ADAR2-DD(E488Q, N496F) was engineered and used to edit a GAC motif in the RAB7A transcript. Values represent mean ± SEM (n = 3). (**b**) 2D histograms comparing the transcriptome-wide A-to-G editing yields observed with full-length and split-ADAR2-DD(E488Q, N496F) constructs. Blue circles indicate the intended target U<u>A</u>G site within the RAB7A transcript. (**c**) A split-RESCUE was engineered and assayed for cytosine to uracil (C-to-U) editing of the RAB7A transcript. Values represent

*Figure 4 continued on next page*

*Figure 4 continued*

mean ± SEM (n = 3), quantified by NGS. (**d**) 2D histograms comparing the transcriptome-wide A-to-G and C-to-U editing yields observed with full-length and split RESCUE constructs. Blue circles indicate the intended target C site within the RAB7A transcript. All experiments were carried out in HEK293FT cells.

The online version of this article includes the following source data for figure 4:

**Source data 1.** Optimizing and expanding the utility of split-ADAR2 deaminase domains.

## Materials and methods
### DMS and screen
#### Oligonucleotide pools
To create the library of single amino acid substitutions in the ADAR2-DD, we ordered an oligonucleotide chip (CustomArray) consisting of six oligonucleotide pools (each 168 bp in length). These pools, in combination, spanned residues 340–600 of the ADAR2-DD. Each of these pools was amplified in a 50 µl PCR using Kapa HiFi HotStart PCR Mix (Kapa Biosystems), 40 ng of synthesized oligonucleotide as template and pool-specific primers. The six PCR products were purified using the QIAquick PCR Purification Kit (Qiagen) to eliminate byproducts.

#### Creation of vectors for cloning oligonucleotide pools

We ordered a gene block (IDT) for MCP-ADAR2-DD-NES and used mutagenesis PCR to create the MCP-ADAR2-DD(E488Q)-NES. These fragments were then used as templates to generate six PCR fragments from which deletions of the MCP-ADAR2-DD-NES and the MCP-ADAR2-DD(E488Q)-NES were created. The deleted regions corresponded to the sequence covered by each of the six oligonucleotide pools and were replaced instead with an Esp3I digestion site. To create the plasmid library, we began by mutating the two Esp3I digestion sites in the LentiCRISPR v2 plasmid (gift from Feng Zhang, Addgene #52961) (*Sanjana et al., 2014*) using PCR mutagenesis followed by Gibson Assembly. Next, we created six cloning vectors for the MCP-ADAR2-DD-NES and MCP-ADAR2-DD(E488Q)-NES, cloning the PCR fragments generated above into the LentiCRISPR v2 vector digested with BamHI and XbaI using Gibson Assembly. All PCRs in this section were carried out using Kapa HiFi HotStart PCR Mix (Kapa Biosystems), 20 ng template and appropriate primers in 20 µl reactions. All digestions in this section were carried out in 50 µl reactions for 3 hr at 37°C using 2 µg of plasmid and 10 units of enzyme(s). All Gibson Assembly reactions in this section were carried out using 50 ng backbone and 30 ng of insert in a 10 µl volume and incubated at 50°C for 1 hr. Digestions and PCRs were purified using the QIAquick PCR Purification Kit (Qiagen).

#### Creation of plasmid library

Once we had six cloning vectors corresponding to the MCP-ADAR2-DD-NES ready, we digested these with Esp3I. These digestions were carried out in 50 µl reactions for 6 hr at 37°C using 2 µg of plasmid and 10 units of enzyme followed by heat inactivation at 65°C for 20 min. The digestion reaction was then purified using the QIAquick PCR Purification Kit (Qiagen). This was followed by cloning of the six oligonucleotide pools into their respective cloning vectors via Gibson Assembly using 50 ng of the digested backbone and 10 ng of the purified oligonucleotide PCR products in a 10 µl reaction, incubated at 50°C for 80 min. The Gibson Assembly reaction was purified by dialysis and used to electroporate ElectroMAX Stbl4 cells (Thermo Fisher) as per the manufacturer's instructions. A small fraction (1–10 µl) of cultures was spread on carbenicillin LB plates to calculate the library coverage, and the rest of the cultures were amplified overnight in 150 ml LB medium containing carbenicillin. A library coverage of at least 400× was ensured before proceeding. Plasmid libraries were sequenced using the MiSeq (300 bp PE run).

## Creation of MS2-adRNA vectors

We began by replacing the Cas9-P2A-Puromycin from the LentiCRISPR v2 with a mCherry-P2A-Hygromycin by digesting the backbone with XbaI and PmeI. We used fusion PCRs to create the mCherry-P2A-Hygromycin-WPRE-3'LTR(Delta U3) insert which was then cloned into the digested backbone via Gibson Assembly. We used PCRs to create a MS2-adRNA-mU6-MS2-adRNA cassette which was cloned into the Esp3I digested backbone via Gibson Assembly. Four vectors with 2× MS2-adRNAs were created targeting 5' and 3' TAG and GAC. All PCRs in this section were carried out using Kapa HiFi HotStart PCR Mix (Kapa Biosystems) in 20 µl reactions. All digestions in the section were carried out in 50 µl reactions for 3 hr at 37°C using 2 µg of plasmid and 10 units of enzymes. All Gibson Assembly reactions in this section were carried out using 50 ng backbone and 20–40 ng of insert in a 10 µl volume and incubated at 50°C for 1 hr. Digestions and PCRs were purified using the QIAquick PCR Purification Kit (Qiagen).

## Lentivirus production

HEK293FT (Thermo Fisher) cells were maintained in DMEM supplemented with 10% FBS (Thermo Fisher) and 1% Antibiotic-Antimycotic (Thermo Fisher) in an incubator at 37°C and 5% $CO_2$ atmosphere. These cells were authenticated by STR and tested for mycoplasma contamination by the vendor. To produce lentivirus particles, HEK293FT cells were seeded in 15 cm tissue culture dishes 1 day before transfection and were 60% confluent at the time of transfection. Before transfection, the culture medium was changed to prewarmed DMEM supplemented with 10% FBS. For each 15 cm dish, 36 µl of Lipofectamine 2000 (Thermo Fisher) was diluted in 1.2 ml OptiMEM (Thermo Fisher). Separately, 3 µg pMD2.G (gift from Didier Trono, Addgene #12259), 12 µg of pCMV delta R8.2 (gift from Didier Trono, Addgene #12263), and 9 µg of lentiviral vector were diluted in 1.2 ml OptiMEM. After incubation for 5 min, the Lipofectamine 2000 mixture and DNA mixture were combined and incubated at room temperature for 30 min. The mixture was then added dropwise to HEK293FT cells. Viral particles were harvested 48 and 72 hr after transfection, further concentrated to a final volume of 500–1000 µl using 100 kDA filters (Millipore), divided into aliquots and frozen at −80°C. Lentivirus was produced individually for all MS2-adRNA vectors and in a pooled format for the libraries. While producing lentivirus, libraries were grouped together as 1 + 2, 3, 4, 5 + 6 so as to facilitate sequencing using the NovaSeq 6000 (250 bp PE run).

## Creation of a clonal cell line with MS2-adRNA

HEK293FT cells grown in a six-well plate were transduced with lentiviruses (high MOI) carrying 2× MS2-adRNA targeting 5' and 3' TAG and GAC to create four different cell lines. For transductions, the lentivirus was mixed with DMEM supplemented with 10% FBS (Thermo Fisher) and Polybrene Transfection reagent (Millipore) at a concentration of 5 µg/ml and added to HEK293FT cells at 40–50% confluency. Hygromycin (Thermo Fisher) was added to the media at a concentration of 100 µg/ml, 48 hr post transduction. Top 1% of mCherry expressing cells for each line were then sorted into a 96-well plate. Three clones of each of the four cell lines were then frozen down.

## Screen

Lentiviral libraries 1 + 2 and 3 were used to transduce clones with the 5' TAG and GAC MS2-adRNA and libraries 4 and 5 + 6 were used to transduce clones with the 3' TAG and GAC MS2-adRNA stably integrated. Transductions were carried out in duplicates. The lentiviral libraries were mixed with DMEM supplemented with 10% FBS (Thermo Fisher), Hygromycin (Thermo Fisher) at 100 µg/ml, Polybrene Transfection reagent (Millipore) at a concentration of 5 µg/ml and added to the stable clones harboring the MS2-adRNA in a 15 cm dish at 40–50% confluency. To ensure most cells received 0 or 1 ADAR2 variant, cells were transduced at a low MOI of 0.2–0.4. 24 hr post transfections, cells were passaged 1:4 into a new 15 cm dish and grown in DMEM supplemented with 10% FBS (Thermo Fisher) and Hygromycin (Thermo Fisher) at 100 µg/ml. Forty-eight hours post transductions, the growth

medium was changed to DMEM supplemented with 10% FBS (Thermo Fisher) and Puromycin (Thermo Fisher) at 3 µg/ml. Seventy-two hours post transduction, fresh growth medium with Puromycin was added to the cells. Ninety-six hours post transductions, the growth media was taken off and cells were washed with PBS and then harvested. Cell pellets were stored at –80°C until RNA extraction. At least 1000× coverage was maintained at all steps of the screen.

## RNA, cDNA, amplifications, indexing

RNA was extracted using the RNeasy mini kit (Qiagen) as per the manufacturer's instructions. cDNA was synthesized from RNA using the Protoscript II First Strand cDNA synthesis Kit (NEB). To ensure library coverage of 500×, 5 ng of RNA was converted to cDNA per library element in every sample of the screen. The volume of each cDNA reaction was 90 µl with 4.5 µg RNA, 45 µl of the reaction mix, 9 µl random primers, and 9 µl enzyme. Samples were incubated in a thermocycler at 25°C for 5 min; 42°C for 80 min; 80°C for 5 min. The entire volume of the cDNA reaction was used to set up PCRs. The volume of each PCR was 100 µl with 44 µl cDNA, 6 µl primers (10 µM), and 50 µl Q5 high fidelity master mix (NEB). The thermocycling parameters were: 98°C for 30 s; 24–28 cycles of 98°C for 10 s, 62°C for 15 s, and 72°C for 35 s; and 72°C for 2 min. The numbers of cycles were tested to ensure that they fell within the linear phase of amplification. The amplicons were 440–570 bp in length and purified using the QIAquick PCR Purification Kit (Qiagen). To continue maintaining at least 500× coverage, at minimum 0.15 ng of the PCR product per library element was used to set up a second PCR adding indices onto the libraries. This was done in 50 µl reactions using 3 µl dual index primers (NEB), 135 ng purified PCR product from the previous reaction and 25 µl Q5 high fidelity master mix (NEB). The thermocycling parameters were: 98°C for 30 s; 5–8 cycles of 98°C for 10 s, 65°C for 20 s, 72°C for 35 s, and 72°C for 2 min. The numbers of cycles were tested to ensure that they fell within the linear phase of amplification. Amplicons were purified with Agencourt AMPure XP beads (Beckman Coulter) at a 0.8 ratio. The libraries were quantified using the Qubit dsDNA HS assay kit (Thermo Fisher) and pooled together at a concentration of 10 nM for sequencing on a 250 bp PE run on the NovaSeq 6000.

## Sequencing analysis

Raw fastq reads were aligned to the ADAR2 reference sequence using minimap2 (*Li, 2018*) in short-read mode with default parameters. For libraries with overlapping paired-end reads, the reads were first combined using FLASH (*Magoč and Salzberg, 2011*). The aligned reads were then classified into library members using strict filtering, that is, reads were only included if they perfectly matched exactly one library member, aside from the target ADAR editing site. The editing rate at this target site was then quantified for each library member and averaged across two replicates with weights for differential coverage. To analyze the degree to which each library member differed in editing rate from the wild type, we performed a two-proportion Z-test using a pooled sample proportion to calculate the standard error of the sampling distribution, and a two-tailed procedure to calculate p-values. Note that the wild-type rate was restricted to the rate measured within each library, such that each library member was compared only to the wild-type rate measured in the same biological context. Z-scores were calculated as follows, where x is the RNA editing rate and n is the number of counts:

$$\underline{x} = \frac{x_{wt}n_{wt}+x_i n_i}{n_{wt}+n_i}$$

$$SE = \sqrt{\underline{x}\left(1-\underline{x}\right)\left(\left(\frac{1}{n_i}\right)+\left(\frac{1}{n_{wt}}\right)\right)}$$

$$Z_i = \frac{x_i - x_{wt}}{SE}$$

Post library classification and editing quantification heatmap plot-ting was done with modified code from Enrich2 (https://github.com/FowlerLab/Enrich2; *Rubin, 2021*; *Rubin et al., 2017*).

## Cloning individual mutants

We began by creating a cloning vector with the MCP inserted into the LentiCRISPR v2 vector digested with BamHI and XbaI using Gibson Assembly. This vector was then digested with BamHI to clone the DD mutants. All mutants were created using mutagenesis PCR followed by Gibson Assembly. All PCRs in this section were carried out using Q5 PCR Mix (NEB), 5 ng template and appropriate primers in 20 µl reactions. All digestions in this section were carried out in 50 µl reactions for 3 hr at 37°C using 3 µg of plasmid and 20 units of enzyme(s). All Gibson Assembly reactions in this section were carried out using 30 ng backbone and 15 ng of insert in a 6 µl volume and incubated at 50°C for 1 hr. Digestions and PCRs were purified using the QIAquick PCR Purification Kit (Qiagen).

## Luciferase assay

All HEK293FT cells were grown in DMEM supplemented with 10% FBS and 1% Antibiotic-Antimycotic (Thermo Fisher) in an incubator at 37°C and 5% $CO_2$ atmosphere. All in vitro luciferase experiments for DMS validations were carried out in HEK293FT cells seeded in 96-well plates, at 25–30% confluency, using 250 ng total plasmid and 0.5 µl of commercial transfection reagent Lipofectamine 2000 (Thermo Fisher). Specifically, every well received 100 ng of the Cluc-W85X(TAG) or Cluc-W85X(TGA) reporters, 50 ng of MCP-ADAR2-DD mutants, and 100 ng of the MS2-adRNA plasmids. In cases where less than three plasmids were needed, a balancing plasmid was added to keep the total amount per well as 250 ng. Forty-eight hours post transfections, 20 µl of supernatant from cells was added to a Costar black 96-well plate (Corning). For the readout, 50 µl of Cypridina Assay buffer was mixed with 0.5 µl Vargulin substrate (Thermo Fisher) respectively and added to the 96-well plate in the dark. The luminescence was read within 10 min on Spectramax i3x or iD3 plate readers (Molecular Devices) with the following settings: 5 s mix before read, 5 s integration time, 1 mm read height.

## RNA editing

RNA editing experiments for targeting 5'-GA-3' were carried out in HEK 293 FT cells seeded in 24-well plates using 1000 ng total plasmid and 2 µl of commercial transfection reagent Lipofectamine 2000 (Thermo Fisher). Specifically, every well received 500 ng each of the MCP-ADAR2-DD variant and the adRNA plasmids. Cells were transfected at 25–30% confluence and harvested 48 hr post transfection for quantification of editing. RNA from cells was extracted using the RNeasy Mini Kit (Qiagen). cDNA was synthesized from 500 ng RNA using the Protoscript II First Strand cDNA synthesis Kit (NEB). One µl of cDNA was amplified by PCR with primers that amplify about 200 bp surrounding the sites of interest using OneTaq PCR Mix (NEB). The numbers of cycles were tested to ensure that they fell within the linear phase of amplification. PCR products were purified using a PCR Purification Kit (Qiagen) and sent out for Sanger sequencing. The RNA editing efficiency was quantified using the ratio of peak heights G/(A + G). For a comprehensive analysis of the ADAR2-DD(E488Q, N496F) mutant, several adenosines within the RAB7A 3'UTR and GAPDH CDS were targeted. The editing efficiencies for each of these adenosines were normalized with that of the ADAR2-DD(E488Q) so as to represent all of the 16 motifs on a single heatmap.

## Split-ADAR2
### Vector design and construction

We began by digesting the pAAV_hU6_mU6_CMV_GFP with AflII to clone the NES-FLAG-MCP-linker and linker-4x λ N-HA-NES downstream of the CMV promoter which were amplified from the MCP-ADAR2-DD-NLS (*Katrekar et al., 2019*) and 4x- λ N-cdADAR2 (*Montiel-Gonzalez et al., 2013*) respectively. AvrII digestion sites were included downstream of the NES-FLAG-MCP-linker and upstream of the linker-4x λ N-HA-NES to facilitate cloning of the split fragments. All split fragments were amplified from the MCP-ADAR2-DD-NLS or MCP-ADAR2-DD(E488Q)-NLS (*Katrekar et al., 2019*). For each split-ADAR2 pair, the N-terminal DD fragment was cloned downstream of the NES-FLAG-MCP-linker and the C-terminal DD fragment was cloned upstream of the linker-4x λ N-HA-NES using Gibson Assembly. MS2-MS2, MS2-BoxB, BoxB-MS2, and BoxB-BoxB adRNA were created by

annealing primers and cloned downstream of the hU6 promoter into the AgeI+ NheI digested pAAV_hU6_mU6_CMV_GFP using Gibson Assembly. All PCRs in this section were carried out using Kapa HiFi HotStart PCR Mix (Kapa Biosystems) in 20 µl reactions. All digestions in this section were carried out in 50 µl reactions for 3 hr at 37°C using 3 µg of plasmid and 20 units of enzyme(s). All Gibson Assembly reactions in this section were carried out using 40 ng backbone and 5–20 ng of insert in a 10 µl volume and incubated at 50°C for 1 hr. Digestions and PCRs were purified using the QIAquick PCR Purification Kit (Qiagen).

### Luciferase assay

All HEK293FT cells were grown in DMEM supplemented with 10% FBS and 1% Antibiotic-Antimycotic (Thermo Fisher) in an incubator at 37°C and 5% $CO_2$ atmosphere. All in vitro luciferase experiments for the split-ADAR2 were carried out in HEK293FT cells seeded in 96-well plates, at 25–30% confluency, using 400 ng total plasmid and 0.6 µl of commercial transfection reagent Lipofectamine 2000 (Thermo Fisher). Specifically, every well received 100 ng each of the Cluc-W85X(TAG) reporter, N- and C-terminal ADAR2 fragments and the adRNA plasmids. In cases where less than four plasmids were needed, a balancing plasmid was added to keep the total amount per well as 400 ng. Forty-eight hours post transfections, 20 µl of supernatant from cells was added to a Costar black 96-well plate (Corning). For the readout, 50 µl of Cypridina Glow Assay buffer was mixed with 0.5 µl Vargulin substrate (Thermo Fisher) and added to the 96-well plate in the dark. The luminescence was read within 10 min on Spectramax i3x or iD3 plate readers (Molecular Devices) with the following settings: 5 s mix before read, 5 s integration time, 1 mm read height.

### RNA editing

All in vitro RNA editing experiments were carried out in HEK293FT cells seeded in 24-well plates using 1500 ng total plasmid and 2 µl of commercial transfection reagent Lipofectamine 2000 (Thermo Fisher). Specifically, every well received 500 ng each of the N- and C-terminal ADAR2 fragments and the adRNA plasmids. In cases where less than three plasmids were needed, a balancing plasmid was added to keep the total amount per well as 1500 ng. Cells were transfected at 25–30% confluence and harvested 48 hr post transfection for quantification of editing. RNA from cells was extracted using the RNeasy Mini Kit (Qiagen). cDNA was synthesized from 500 ng RNA using the Protoscript II First Strand cDNA synthesis Kit (NEB). One µl of cDNA was amplified by PCR with primers that amplify about 200 bp surrounding the sites of interest using OneTaq PCR Mix (NEB). The numbers of cycles were tested to ensure that they fell within the linear phase of amplification. PCR products were purified using a PCR Purification Kit (Qiagen) and sent out for Sanger sequencing. The RNA editing efficiency was quantified using the ratio of peak heights $G/(A + G)$. RNA-seq libraries were prepared from 250 ng of RNA, using the NEBNext Poly(A) mRNA magnetic isolation module and NEBNext Ultra RNA Library Prep Kit for Illumina. Samples were pooled and loaded on an Illumina Novaseq 6000 (100 bp paired-end run) to obtain 40–45 million reads per sample.

### Quantification of RNA-seq A-to-G editing

RNA-seq analysis for quantification of transcriptome-wide A-to-G editing was carried out as described in *Katrekar et al., 2019*.

## Acknowledgements

We thank members of the Mali lab for discussions, advice and help with experiments. This work was generously supported by UCSD Institutional Funds, NIH grants (R01HG009285, R01CA222826, R01GM123313, 1K01DK119687), and Department of Defense Grant (DOD PR210085). This publication includes data generated at the UC San Diego IGM Genomics Center utilizing an Illumina NovaSeq 6000 that was purchased with funding from a National Institutes of Health SIG grant (#S10 OD026929).

## Additional information

### Competing interests

Dhruva Katrekar, Yichen Xiang: has filed a patent pertaining to the screening methodology, novel mutants and splitting of the ADAR2-DD (application number: 63/075,717). Is now an employee of Shape Therapeutics. The other authors declare that no competing interests exist.

### Funding

| Funder | Grant reference number | Author |
|---|---|---|
| National Human Genome Research Institute | R01HG009285 | Prashant Mali |
| National Cancer Institute | R01CA222826 | Prashant Mali |
| National Institute of General Medical Sciences | R01GM123313 | Prashant Mali |
| U.S. Department of Defense | PR210085 | Prashant Mali |

The funders had no role in study design, data collection and interpretation, or the decision to submit the work for publication.

### Author contributions

Dhruva Katrekar, Conceptualization, Data curation, Formal analysis, Investigation, Methodology, Writing – original draft, Writing – review and editing; Yichen Xiang, Anushka Saha, Investigation; Nathan Palmer, Dario Meluzzi, Formal analysis; Prashant Mali, Conceptualization, Funding acquisition, Investigation, Methodology, Project administration, Supervision, Writing – original draft, Writing – review and editing

### Author ORCIDs

Dhruva Katrekar http://orcid.org/0000-0002-8028-3244
Nathan Palmer http://orcid.org/0000-0001-6347-9379
Prashant Mali http://orcid.org/0000-0002-3383-1287

### Decision letter and Author response

Decision letter https://doi.org/10.7554/eLife.75555.sa1
Author response https://doi.org/10.7554/eLife.75555.sa2

## Additional files

### Supplementary files

- Supplementary file 1. Off-target characterization of the split-ADAR2 constructs.
- Transparent reporting form

### Data availability

Sequencing data will be accessible via NCBI GEO under accession GSE158656. Source data has been made available with the submission.

The following dataset was generated:

| Author(s) | Year | Dataset title | Dataset URL | Database and Identifier |
|---|---|---|---|---|
| Katrekar D, Meluzzi D, Palmer N, Mali P | 2022 | Comprehensive interrogation of the ADAR2 deaminase domain for engineering enhanced RNA base-editing activity, functionality and specificity | https://www.ncbi.nlm.nih.gov/geo/query/acc.cgi?acc=GSE158656 | NCBI Gene Expression Omnibus, GSE158656 |

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
