## [Editor Report]

This manuscript provides a deep mutational scanning of the deaminase domain of human ADAR2 to provide a comprehensive assessment of amino acids that alter editing activity at a specific adenosine flanked by preferred nucleotides (UAG). The results are quite important in terms of impact on precision medicine.

---

## [Decision Letter]

**Decision letter after peer review:**

[Editors’ note: the authors submitted for reconsideration following the decision after peer review. What follows is the decision letter after the first round of review.]

Thank you for submitting your work entitled "Comprehensive interrogation of the ADAR2 deaminase domain for engineering enhanced RNA editing activity and specificity" for consideration by *eLife*. Your article has been reviewed by 3 peer reviewers, and the evaluation has been overseen by a Reviewing Editor and a Senior Editor. The following individuals involved in review of your submission have agreed to reveal their identity: Timothy A Whitehead (Reviewer #2); Nina Papavasiliou (Reviewer #3).

Our decision has been reached after consultation between the reviewers. Based on these discussions and the individual reviews below, we regret to inform you that your work will not be considered further for publication in *eLife*.

All three reviewers raised substantive concerns regarding description of the methods and statistical analyses. Given these concerns, we had no choice but to decline the paper, but encourage you to resubmit if and when you can address the points raised by the referees as thoroughly as possible. The issues are particularly concerning since this manuscript was submitted as a research method or tool paper.

*Reviewer #1:*

The authors aims to enhance efficiency and specificity of RNA engineering mediated by ADAR2 deaminase domain (ADAR2-DD) by three approaches. First, they interrogated the residues in a portion of ADAR2-DD by deep mutational scanning (DMS), providing comprehensive map of residues important for the core region of ADAR2-DD. Second, they identified a novel mutation N496F that increases editing efficiency for a 5'-GA-3' motif. Third, they showed the split ADAR2-DD design with good editing efficiency and specificity. This work provides deeper understanding of ADAR2 deaminase domain as a tool for RNA directed engineering. Although the work presents advances, I have several concerns that the authors need to address in more detailed analysis.

1) The DMS screen described in Figure 1a is not intuitive to understand. Where exactly are the editing sites in the ADAR2-DD? How many sites are there in total? How are the editing sites and mutations identified in the sequencing reads? A better drawing, explanation, figure legend and methods are needed.

2) For the DMS method, since the editing site is in ADAR2-DD itself, when making mutations the structure of the RNA substrate might change, especially the ones near the editing site. Then the effect might not be sole consequence of the protein mutant. If RNA substrate might change in structure, is the DMS result still consistent with the validation result using the cluc assays?

3) In addition to measuring the editing level of targeted sites, can the authors measure off target in the deep sequencing data? This would result in both editing efficiency and editing specificity measurement.

4) ADAR1 is perhaps expressed in the chosen human cells. Would any of the results from the DMS and the validation be complicated by the editing from endogenous ADAR1?

5) It is very interesting that many mutants showed equal or higher activity than the well-known E488Q mutant. What are the efficiency and specificity for these (at least for some representative ones from the validation)?

6) For the N496F mutant, what are the transcriptome wide off-targeting data? What about other non-GA sequences?

7) How was the split ADAR2-DD chosen? Current writing is very simple. It would be useful for the community if the authors provide more details of the reasoning.

8) What is the correlation of the luciferase assay signal to the actual editing level of the transcript? What if comparing all designs (mutations and split-ADAR2-DD) in the same assay so we can see direction comparison of the editing efficiency and specificity? Preferably, using editing level of a target site as a readout to compare all designs and the off-target analysis by RNA-seq.

9) To validate all the findings in this work, it would be desired to show how an engineered ADAR2 DD, in a split fashion, would edit an endogenous substrate with a non-UAG motif (such as GAC). What would be the editing efficiency (% editing level) and the transcriptome-wide specificity?

*Reviewer #2:*

Katrekar and colleagues developed a screen for deaminase acting on RNA (ADAR) and screened most single amino acid substitutions across the catalytically active domain for RNA editing and for activity at 5'-GA-3' motifs. Separately, they developed a split ADAR and evaluated specific and off-target RNA editing using whole transcriptomes. The paper does not read like a coherent story and instead is two separate papers: Figure 1 – 2 involve the screen and evaluation of single clones resulting from the screen, whereas Figure 3-4 involve the split ADAR. The strengths of the paper involve the novelty of the genetic screen and, separately, the development and validation of the split ADAR system. There exist major concerns about the representation of the results from the screen, along with minor suggestions on the split ADAR story.

1. The statistical underpinning of the validity of the screening results are unexplored in the main text and need to be described accurately. The authors split between Z scores (Figure 1), Fold change in DMS relatiive to ADAR2-DD (Figure 2), % edited (in supporting information files and Figure 1d), DMS log2 fold change (SI Figure 2). Each of these screening outputs (if all are included) need to be described in the main text and justified. My personal opinion is that one or at most two metrics can be used in the paper to avoid confusion. I have particular concerns in this section about the following:

a. Replicates for the screen. The paper only lists replicates in three places, and in no place was how this replicate performed. How were they performed? Biological replicates? Technical replicates? Different days? These experimental details need to be discussed explicitly.

b. Of concern for the replicates are the relatively low correlation between replicates (R^2^ = 0.48 by my calculation). The correlation is not discussed at all in the main text – this data needs to be explicit for the reader to judge for herself the validity of the data presented.

c. The replicate showed in SI Figure 1d has a correlation missing for the worst performing sample ("wt-X-TAG") and the meaning for wt-X, wt-Y, etc are not described.

d. The validation performed isogenically involves cherrypicked samples with low variance between them (R^2^ for the variants described in figure 2b are 0.87) and don't represent a fair comparison. The authors state that "We observed that a majority of the mutants (85%) followed the same trend in our arrayed validation as seen in the pooled screens" but the meaning behind the sentence is not clear. What does the same trend mean and how is it calculated? Determine the statistical significance using a t test and show comparisons between isogenic datasets using rank correlation or R^2^ correlation.

e. Points a-d lead to the following conclusion that the screen, while clever and well implemented, has relatively high error and the data should not be presented as a heat map as the authors present in Figure 1. Deep mutational scanning experiments where data is presented as heat maps typically have R^2^ values of 0.8 or higher. This data is useful – the data for conservation at each position should be relatively robust even with the error in the screen reported in the paper. This screen can also be used to identify 'hits'.

*Reviewer #3:*

The paper by Mali and colleagues is an interesting mix of experiments on ADAR2 functionality: on mutations that increase activity, on a split ADAR2 construct appears to decrease off target effects, and on a split RESCUE construct that is said to also increase the specificity of C to U editing.

The notion of a split ADAR is certainly novel (brought together by the binding in situ of two elements on the RNA – an MS2 and boxB element, plus a second pair). However the paper would really benefit from being more explicit on some of the results. For instance, the "decrease of off target effects" though apparently significant, would benefit from some nuance – for example are there commonalities to the off-target targets? (are there RAB7A-specific off-targets vs KRAS specific ones and what would that imply?) In other words how generalizable to other transcripts are these findings?

Continuing with the notion of tradeoffs, are GAC/GAG-focused mutants "worse" on other triplets? and which?

Finally, given how little has been published on targeted C to U editing (excepting RESCUE), it is important to treat figure 4d as a little more than an afterthought – with a comprehensive analysis equivalent to the treatment of A to I editing.

[Editors’ note: further revisions were suggested prior to acceptance, as described below.]

Thank you for resubmitting your work entitled "Comprehensive interrogation of the ADAR2 deaminase domain for engineering enhanced RNA editing activity and specificity" for further consideration by *eLife*. Your revised article has been evaluated by James Manley (Senior Editor) and Timothy Nilsen (Reviewing Editor).

The manuscript has been improved but there are some remaining issues that need to be addressed, as outlined below:

While both reviewers were quite positive about your revised paper, they felt that additional textual changes and/or additions as outlined below would further improve the manuscript. Please address these comments as thoroughly as possible.

*Reviewer #1:*

The manuscript by Mali and colleagues is substantially improved from the earlier version. If I had a single comment to make on the revision (and only because they now have the data to look) is this. Normally, a guide RNA bound to the coding region (as will be the case for the cAg in Rab7a) would be expected to reduce transcript abundance (through RNAi like effects). In view of the data in figure 4a/4b, is this true? if so it would be important to point out, because this is not normally something that would be observed in a "restore of a stop codon" situation, but it is something we need to worry about in terms of therapeutic efficiency.

*Reviewer #2:*

This revised manuscript provides a deep mutational scanning of the deaminase domain of human ADAR2 to provide a comprehensive assessment of amino acids that alter editing activity at a specific adenosine flanked by preferred nucleotides (UAG). The author recover 33 individual mutations that either increase or decrease editing, including several mutations that were previously known to impact editing. The authors perform a second DMS starting with a known hyperactive mutant and seeking to obtain a mutant that has altered preferences for the nearest neighbors of the target adenosine. This second goal is quite important in terms of impact on precision medicine. The last goal of the paper is to develop a split ADAR method of editing target adenosines with the goal of reducing off target adenosines, again an important technological advancement for therapeutic use of ADARs. Overall, the revisions adequately address the concerns about clarity, experimental approaches and statistical analysis.

Two areas that need to be addressed are listed below.

1. It would be beneficial if the authors specifically identified the novel mutations identified in the deaminase domain determined from the initial DMS experiment on the UAG codon (SI Figure 2). Furthermore, the initial reviews noted that there were several mutations (ex. D419W, D362R, D365R, etc) that exhibit a similar elevated activity as the well-described E488Q hyperactive mutant on the UAG substrate. The authors were asked (by reviewer 1) whether these hyperactive mutants are specific to UAG or also behave similar to E488Q (exhibit increased editing at less preferred codons). This was not addressed in the revised manuscript.

2. The second DMS screen identified only one mutant, N496F, that could significantly enhance editing of a GAC codon. The authors did not recover this mutant in the initial screen, is that due to the lack of N496F affecting editing at UAG codons?

The abstract describes this mutant as "greatly increased enzymatic activity at 5' GAN-3' motifs". This language is overstated both in terms of the activity (which is simply 1.1-2 fold enhanced (Figure 1h)) and with regards to the specific motif. The comprehensive assessment of the specificity of this mutant (requested in the initial review, SI Figure 3c) indicates this mutant has enhanced activity not only for GAN motifs but also CAN motifs, with CAC being the second most edited codon after GAA (and above several other GAN codons).

The authors should both tone down the language in the abstract and discuss the lack of specificity of the E448Q, N486F mutant, especially in terms of what may occur with off-targets. The authors have already performed the experiment (Figure 3b and Figure 4b) but do not discuss the data in this regard, despite the initial request by Reviewer 1. This is particularly important as the authors stress that finding mutants with altered preference is important for precision medicine, but if these ADAR mutants also increase off-target editing, the findings are less exciting-albeit more rationale for using the split ADAR technology developed.

The methodology for the comprehensive analysis of editing preferences should be added to the manuscript.

---

## [Author Response]

[Editors’ note: the authors resubmitted a revised version of the paper for consideration. What follows is the authors’ response to the first round of review.]

Reviewer #1:The authors aims to enhance efficiency and specificity of RNA engineering mediated by ADAR2 deaminase domain (ADAR2-DD) by three approaches. First, they interrogated the residues in a portion of ADAR2-DD by deep mutational scanning (DMS), providing comprehensive map of residues important for the core region of ADAR2-DD. Second, they identified a novel mutation N496F that increases editing efficiency for a 5'-GA-3' motif. Third, they showed the split ADAR2-DD design with good editing efficiency and specificity. This work provides deeper understanding of ADAR2 deaminase domain as a tool for RNA directed engineering. Although the work presents advances, I have several concerns that the authors need to address in more detailed analysis.(1) The DMS screen described in Figure 1a is not intuitive to understand. Where exactly are the editing sites in the ADAR2-DD? How many sites are there in total? How are the editing sites and mutations identified in the sequencing reads? A better drawing, explanation, figure legend and methods are needed.

To improve clarity we have now split Figure 1 into Figures 1a, 1b and 1c. The novelty of the DMS lies in coupling RNA editing and mutant information by linking them on the same transcript, thereby enabling direct measurement of biochemical activity (vs. via a surrogate assay). To enable this the editing sites are located in the ADAR2-deaminase domain (DD) outside of the region where single amino acid substitutions are created (residues 340-600). Specifically, we assayed UAG and GAC editing yields for ADAR2-DD mutants between amino acids positions 340-468 via sites located at amino acid positions 332 and 329. To not alter the protein coding sequence, at these sites an A-to-G substitution creates a synonymous amino acid change. Similarly, we assayed UAG and GAC editing yields for ADAR2-DD mutants between amino acids positions 469-600 via sites located at amino acid positions 621 and 626. To not alter the protein coding sequence, at these sites an A-to-G substitution creates a synonymous amino acid change. By transducing cells at a low multiplicity of infection (MOI), we ensure that each cell receives a single mutant. The schematic for the location of editing sites is highlighted in SI Figure 1a. When corresponding transcription and translation occur in a cell, the MCP-ADAR2 mutant complexes with the MS2 guide RNA to edit its own transcript. If a mutant is highly active, a higher number of the mutant transcripts will contain the A-to-G substitution at the editing site, as compared to a less functional mutant. By isolating RNA, converting it to cDNA and amplifying the ADAR2DD, we could quantify the total number of reads arising from each mutant and determine the fraction of reads that contain the targeted A-to-G edit. UAG and GAC editing yields were assayed in independent experiments via use of ADAR2 or ADAR2 (E488Q) mutant libraries, and in cells harboring distinct MS2 guide RNAs.

(2) For the DMS method, since the editing site is in ADAR2-DD itself, when making mutations the structure of the RNA substrate might change, especially the ones near the editing site. Then the effect might not be sole consequence of the protein mutant. If RNA substrate might change in structure, is the DMS result still consistent with the validation result using the cluc assays?

We understand the reviewer’s concern regarding the local structure of the target being altered by the mutations in the ADAR2-DD sequence. The DMS assay was purposefully designed such that: (1) the editing sites were chosen to be substantially outside of the region where the mutations were made, with the minimum distance between the editing site and mutation being >20 bp; (2) additionally, the mutations were created outside the binding site of the guide RNA; (3) furthermore, we confirmed the effects of these mutations on the overall free energy of RNA folding are predicted to be minimal; and (4) importantly, the DMS results were consistent with the validation results, confirming the efficacy of the screen. See Figure 1-figure supplement 2.

(3) In addition to measuring the editing level of targeted sites, can the authors measure off target in the deep sequencing data? This would result in both editing efficiency and editing specificity measurement.

While we have assayed activity and transcriptome-wide specificity for individual mutants in this study, given the pooled nature of the screen it is however not possible to measure at-scale both on-target the off-target editing associated with each mutant in the library. This would in principle require single cell RNA seq measurements. However these assays in current formats (such as via 10X genomics) yield: (1) very sparse data, and (2) provide sequence information only for RNA regions close to poly-A sites.

(4) ADAR1 is perhaps expressed in the chosen human cells. Would any of the results from the DMS and the validation be complicated by the editing from endogenous ADAR1?

ADAR1 is indeed expressed in the HEK293FT cells in which the experiment was carried out. However, when using the short MS2-adRNA antisense sequences (length 20 bp) there is no recruitment of endogenous ADAR1 (data reproduced below from Katrekar et al. Nature Methods, 2019).

**Author response image 1. sa2fig1:** Specifically, on-target RNA editing by MCP–ADAR2 DD-NLS required co-expression of the MS2 adRNA. GluR2 adRNA and MS2 adRNA used in this experiment had an antisense domain of length 20. Values represent mean ± s.e.m. (n = 3). All experiments were carried out in HEK293T cells.

5) It is very interesting that many mutants showed equal or higher activity than the well-known E488Q mutant. What are the efficiency and specificity for these (at least for some representative ones from the validation)?

We agree and present corresponding detailed characterization data for ADAR2(E488Q, N496F) - notably, this novel mutant showed enhanced activity against GAC motifs which wt-ADAR2 is unable to efficiently edit. We have rigorously characterized this hit via editing and specificity analysis, including all 16 *5’-NAN-3’* motifs, and have additionally also created all-in-one-split versions of the same. See Figures 4a and b, Figure 1-figure supplement 3.

We have also validated several additional mutants identified in the screen via both a luciferase reporter assay and direct RNA editing measurements. See Figure 1—figure supplement 2.

6) For the N496F mutant, what are the transcriptome wide off-targeting data? What about other non-GA sequences?

As suggested by the reviewer, we have now carried out deep RNA-seq to quantify the transcriptome wide off-target editing observed with the ADAR2(E488Q, N496F) double mutant expressed both as a full domain and a split-domain. Per our engineered designs, the latter format eliminates off-target RNA editing while still retaining robust on-target activity. See Figure 4b.

Additionally, we also comprehensively examined editing across all possible 5’ and 3’ flanking nucleotides, and corresponding data comparing the profiles with ADAR2(E488Q), see Figure 1—figure supplement 3c.

7) How was the split ADAR2-DD chosen? Current writing is very simple. It would be useful for the community if the authors provide more details of the reasoning.

We agree, and the rationale for choosing the residues is now more explicitly detailed in the main text: “*Examining the results of the DMS (focusing on sites with high mutability), as well as the crystal structure of the ADAR2-DD (focusing on high solvent accessible surface area), and residue conservation scores across species (focusing on low scores of conservation), we identified 18 putative regions for splitting the protein.*”

8) What is the correlation of the luciferase assay signal to the actual editing level of the transcript? What if comparing all designs (mutations and split-ADAR2-DD) in the same assay so we can see direction comparison of the editing efficiency and specificity? Preferably, using editing level of a target site as a readout to compare all designs and the off-target analysis by RNA-seq.

As suggested, we have now plotted the editing levels of the transcript against the luciferase signal (RLU). The Pearson correlation for the scatter plot is 0.818 while the Spearman correlation is 0.824. See Figure 1—figure supplement 2.

Additionally, as requested, we have also compared the full-length ADAR2, the E488Q mutant and the E488Q, N496F double mutant and their respective all-in-one-split counterparts via RNA-seq (while editing the same UAG site in the RAB7A 3’ UTR). See Figures 3b, 4b.

9) To validate all the findings in this work, it would be desired to show how an engineered ADAR2 DD, in a split fashion, would edit an endogenous substrate with a non-UAG motif (such as GAC). What would be the editing efficiency (% editing level) and the transcriptome-wide specificity?

As suggested by the reviewer, we have now created an all-in-one-split-ADAR2DD(E488Q, N496F) and compared it with the all-in-one-split-ADAR2(E488Q), and indeed can confirm a significant increase in editing efficiency of the GAC motif. Additionally, we show that splitting the enzyme does significantly reduce off-target editing as compared to the full-length MCP-ADAR2-DD(E488Q, N496F). This RNA-seq was carried out in the context of the same UAG editing site of the RAB7A used in all other samples so as to enable a side-by-side comparison between them. See Figures 3b and 4b.

Reviewer #2:Katrekar and colleagues developed a screen for deaminase acting on RNA (ADAR) and screened most single amino acid substitutions across the catalytically active domain for RNA editing and for activity at 5'-GA-3' motifs. Separately, they developed a split ADAR and evaluated specific and off-target RNA editing using whole transcriptomes. The paper does not read like a coherent story and instead is two separate papers: Figure 1 – 2 involve the screen and evaluation of single clones resulting from the screen, whereas Figure 3-4 involve the split ADAR. The strengths of the paper involve the novelty of the genetic screen and, separately, the development and validation of the split ADAR system. There exist major concerns about the representation of the results from the screen, along with minor suggestions on the split ADAR story.1. The statistical underpinning of the validity of the screening results are unexplored in the main text and need to be described accurately. The authors split between Z scores (Figure 1), Fold change in DMS relatiive to ADAR2-DD (Figure 2), % edited (in supporting information files and Figure 1d), DMS log2 fold change (SI Figure 2). Each of these screening outputs (if all are included) need to be described in the main text and justified. My personal opinion is that one or at most two metrics can be used in the paper to avoid confusion.

As suggested by the reviewer, we have now switched to log2 fold change for all the replicate correlations and individual hit validations, and for the screen heatmap used Zscores.

I have particular concerns in this section about the following:a. replicates for the screen. The paper only lists replicates in three places, and in no place was how this replicate performed. How were they performed? Biological replicates? Technical replicates? Different days? These experimental details need to be discussed explicitly.

Here are the specific details (also included in the manuscript): “*Two biological replicates were performed in independent plates of cells transduced with independent vials of lentivirus.”*

b. Of concern for the replicates are the relatively low correlation between replicates (R^2^ = 0.48 by my calculation). The correlation is not discussed at all in the main text – this data needs to be explicit for the reader to judge for herself the validity of the data presented.

We agree. To address this, we have now increased the sequencing depth and also used more stringent filtering. Updated metrics are in Figure 1d and Figure 1-figure supplement 1b.

We have also included a statement regarding the R^2^ value in the main text which now reads: *“The deaminase domain transcripts for each variant also contained the associated A-to-I editing yields, which were then quantified for both replicates of the DMS (R^2^ = 0.687).”*

c. The replicate showed in SI Figure 1d has a correlation missing for the worst performing sample ("wt-X-TAG") and the meaning for wt-X, wt-Y, etc are not described.

This section has been updated.

d. The validation performed isogenically involves cherrypicked samples with low variance between them (R^2^ for the variants described in figure 2b are 0.87) and don't represent a fair comparison. The authors state that "We observed that a majority of the mutants (85%) followed the same trend in our arrayed validation as seen in the pooled screens" but the meaning behind the sentence is not clear. What does the same trend mean and how is it calculated? Determine the statistical significance using a t test and show comparisons between isogenic datasets using rank correlation or R^2^ correlation.

The samples for validations were picked based on Z-score (low, medium, high). We have determined the statistical significance using an independent t-test with unequal variance (Welch’s t-test). 24 out of the 33 mutants that were validated are not significantly different from the screening data (p>0.05). For the 9 samples that are significantly different it is an issue of magnitude rather than direction with the screen tending to overestimate the validation. The Pearson correlation between the arrayed validations and data from the screen is 0.818. The Spearman correlation is 0.824. Taken together, the screen is accurate in predicting whether a particular mutation will impair, improve or not alter RNA editing. See Supplementary Figure 1-figure supplement 2a.

e. Points a-d lead to the following conclusion that the screen, while clever and well implemented, has relatively high error and the data should not be presented as a heat map as the authors present in Figure 1. Deep mutational scanning experiments where data is presented as heat maps typically have R^2^ values of 0.8 or higher. This data is useful – the data for conservation at each position should be relatively robust even with the error in the screen reported in the paper. This screen can also be used to identify 'hits'.

By further increasing the sequencing depth, and also using more stringent filtering, our data now has an R^2^ = 0.687. However, per the reviewer's suggestion, we have now deemphasized the DMS screen, moved the corresponding heat map to the Supplementary Information section, and primarily utilized it in the manuscript as a screen for identifying novel hits. We would however like to highlight that this novel screening format directly measures RNA editing yields and thus enzymatic activity. This is unlike the large majority of mutagenesis screens in literature that rely primarily on surrogate readouts.

Reviewer #3:The paper by Mali and colleagues is an interesting mix of experiments on ADAR2 functionality: on mutations that increase activity, on a split ADAR2 construct appears to decrease off target effects, and on a split RESCUE construct that is said to also increase the specificity of C to U editing.The notion of a split ADAR is certainly novel (brought together by the binding in situ of two elements on the RNA – an MS2 and boxB element, plus a second pair). However the paper would really benefit from being more explicit on some of the results. For instance, the "decrease of off target effects" though apparently significant, would benefit from some nuance – for example are there commonalities to the off-target targets? (are there RAB7A-specific off-targets vs KRAS specific ones and what would that imply?) In other words how generalizable to other transcripts are these findings?

We thank the reviewer for this suggestion. We have now carried out RNA-seq analysis on the KRAS targeting samples. A closer look at the off-targets confirms that they are guide RNA dependent for the split-ADAR system. This is in contrast to the overexpressed full length deaminases where off-target edits are primarily driven by the non-specific dsRNA binding activity of the enzyme. A statement towards this has now also been included in the main text: “A closer look at the off-targets revealed that in case of the splitADAR2 system, highly edited off-targets were guide RNA sequence dependent. This is in contrast to full-length deaminase domain overexpression where off-targets were predominantly deaminase domain driven.”

A summary of full length deaminase domain overexpression is shown in Author response image 2 both in the presence and absence of guide RNA.

1753 off-targets were shared between all constructs. This indicates that enzyme preferences dictate off-targets as 2 of the latter constructs lack a guide RNA.We have also closely examined the off-targets in 4 of our split-deaminase samples. A summary is shown in Author response image 3.

**Author response image 3. sa2fig3:** 

68 off-targets were shared only between the KRAS targeting constructs. 3 of 68 shared off-targets are seen in the PAICS transcript, and an alignment of the reverse complement of the guide and the off-target site is shown:Reverse complement of guide: TCCTCATGTTACAAACTTGTGGTGC

Highly edited off-target : TGCTAGGGTTACAGACATGAGCCAC

Continuing with the notion of tradeoffs, are GAC/GAG-focused mutants "worse" on other triplets? and which?

We agree this is an important characterization of the mutant. We have now looked at editing across other motifs and compared it to the ADAR2(E488Q). See Figure 1-figure supplement 3c.

Finally, given how little has been published on targeted C to U editing (excepting RESCUE), it is important to treat figure 4d as a little more than an afterthought – with a comprehensive analysis equivalent to the treatment of A to I editing.

We fully agree, and as suggested by the reviewer, have now delved deeper into C-to-U editing via RESCUE. Specifically, we carried out RNA-seq analyses which confirmed that splitting the enzyme reduces off-targets both in the A-to-I space as well as the C-to-U space. See Figures 4c and 4d.

[Editors’ note: what follows is the authors’ response to the second round of review.]

Reviewer #1:The manuscript by Mali and colleagues is substantially improved from the earlier version. If I had a single comment to make on the revision (and only because they now have the data to look) is this. Normally, a guide RNA bound to the coding region (as will be the case for the cAg in Rab7a) would be expected to reduce transcript abundance (through RNAi like effects). In view of the data in figure 4a/4b, is this true? if so it would be important to point out, because this is not normally something that would be observed in a "restore of a stop codon" situation, but it is something we need to worry about in terms of therapeutic efficiency.

This is an important point brought up by the reviewer. RNAi like effects could potentially be seen while targeting a CDS. We have now carried out qPCRs on two targets, in the presence or absence of a guide RNA targeting a CAG in the RAB7A transcript and a CAT in the GAPDH transcript. We do not observe any RNAi like effects in presence of the guide RNA.

**Author response image 4. sa2fig4:** 

Reviewer #2:This revised manuscript provides a deep mutational scanning of the deaminase domain of human ADAR2 to provide a comprehensive assessment of amino acids that alter editing activity at a specific adenosine flanked by preferred nucleotides (UAG). The author recover 33 individual mutations that either increase or decrease editing, including several mutations that were previously known to impact editing. The authors perform a second DMS starting with a known hyperactive mutant and seeking to obtain a mutant that has altered preferences for the nearest neighbors of the target adenosine. This second goal is quite important in terms of impact on precision medicine. The last goal of the paper is to develop a split ADAR method of editing target adenosines with the goal of reducing off target adenosines, again an important technological advancement for therapeutic use of ADARs. Overall, the revisions adequately address the concerns about clarity, experimental approaches and statistical analysis.Two areas that need to be addressed are listed below.1. It would be beneficial if the authors specifically identified the novel mutations identified in the deaminase domain determined from the initial DMS experiment on the UAG codon (SI Figure 2). Furthermore, the initial reviews noted that there were several mutations (ex. D419W, D362R, D365R, etc) that exhibit a similar elevated activity as the well-described E488Q hyperactive mutant on the UAG substrate. The authors were asked (by reviewer 1) whether these hyperactive mutants are specific to UAG or also behave similar to E488Q (exhibit increased editing at less preferred codons). This was not addressed in the revised manuscript.

While we do understand the importance of further characterizing the mutants identified in the initial DMS, we want to point out that the goal of this DMS was primarily to create a mutagenesis map of the ADAR2 and thereby understand which residues tolerate mutations and which ones do not. An exhaustive list of mutants with associated RNA editing activities against the UAG is listed in the supporting tables. We have now also evaluated a handful of these mutants against CAG and GAC motifs. Given that the screen was carried out against a UAG motif, the effects of these mutations on editing of CAG and GAC motifs was not as pronounced as the UAG motif.

**Author response image 5. sa2fig5:** 

2. The second DMS screen identified only one mutant, N496F, that could significantly enhance editing of a GAC codon. The authors did not recover this mutant in the initial screen, is that due to the lack of N496F affecting editing at UAG codons?

The second DMS was carried out in the E488Q background while the first was carried out in the wild type ADAR2. The second screen helped uncover a double mutant E488Q, N496F which had enhanced activity against a GAC triplet. When this double mutant was evaluated against a UAG, the improvement in editing was 1.12 fold compared to the E488Q as can be seen in Figure 1—figure supplement 3c. This N496F mutant was on an average 1.2 fold better than the ADAR2 at editing a UAG motif in the two replicates of the first screen. This tells us that the N496F itself does not greatly alter editing at UAG motifs.

The abstract describes this mutant as "greatly increased enzymatic activity at 5' GAN-3' motifs". This language is overstated both in terms of the activity (which is simply 1.1-2 fold enhanced (Figure 1h)) and with regards to the specific motif. The comprehensive assessment of the specificity of this mutant (requested in the initial review, SI Figure 3c) indicates this mutant has enhanced activity not only for GAN motifs but also CAN motifs, with CAC being the second most edited codon after GAA (and above several other GAN codons).

We have now toned down the abstract and the relevant statement reads: “This enabled us to create a domain wide mutagenesis map while also revealing a novel hyperactive variant with improved enzymatic activity at 5’-GAN-3’ motifs.”

5’-CAC-3’ definitely is highly edited, however, this is not the case with other 5’-CAN-3’ motifs but instead with 5’-NAC-3’ motifs. We believe that since the screen was carried out while targeting a 5’-GAC-3’ motif, we have selected for a mutant with improved editing in the context of a 5’-G and a 3’-C. This is now included in the main text and reads: We also confirmed that this new variant was at least as efficient as the E488Q at editing all other motifs, with improved editing also observed at 5’-NAC-3’ motifs.

The authors should both tone down the language in the abstract and discuss the lack of specificity of the E448Q, N486F mutant, especially in terms of what may occur with off-targets. The authors have already performed the experiment (Figure 3b and Figure 4b) but do not discuss the data in this regard, despite the initial request by Reviewer 1. This is particularly important as the authors stress that finding mutants with altered preference is important for precision medicine, but if these ADAR mutants also increase off-target editing, the findings are less exciting-albeit more rationale for using the split ADAR technology developed.

We agree with the reviewer and have made the necessary changes to the manuscript.

We have altered the abstract and the relevant statement reads: However, exogenous delivery of ADAR enzymes, especially hyperactive variants, leads to significant transcriptome wide off-targeting.

We have also included the following statement in the main text: Although the full-length ADAR2-DD(E488Q, N496F) was highly promiscuous, splitting it enabled high transcriptome-wide specificity while targeting a UAG in the RAB7A transcript.

Additionally, we have also added the following statement to the Discussion section: However, like the ADAR2-DD(E488Q), the ADAR2-DD(E488Q, N496F) showed increased bystander editing and transcriptome wide off-targeting as compared to the ADAR2-DD.

The methodology for the comprehensive analysis of editing preferences should be added to the manuscript.

This statement has now been added to the methods section and reads: For a comprehensive analysis of the ADAR2-DD(E488Q, N496F) mutant, several adenosines within the RAB7A 3’UTR and GAPDH CDS were targeted. The editing efficiencies for each of these adenosines were normalized with that of the ADAR2-DD(E488Q) so as to represent all of the 16 motifs on a single heatmap.